# Rehoming and Other Refinements and Replacement in Procedures Using Golden Hamsters in SARS-CoV-2 Vaccine Research

**DOI:** 10.3390/ani13162616

**Published:** 2023-08-14

**Authors:** Malan Štrbenc, Urška Kuhar, Duško Lainšček, Sara Orehek, Brigita Slavec, Uroš Krapež, Tadej Malovrh, Gregor Majdič

**Affiliations:** 1Institute for Preclinical Sciences, Veterinary Faculty, University of Ljubljana, 1000 Ljubljana, Slovenia; gregor.majdic@vf.uni-lj.si; 2Institute for Microbiology and Parasitology, Veterinary Faculty, University of Ljubljana, 1000 Ljubljana, Slovenia; urska.kuhar@vf.uni-lj.si (U.K.); tadej.malovrh@vf.uni-lj.si (T.M.); 3Department of Synthetic Biology and Immunology, National Institute of Chemistry, 1000 Ljubljana, Slovenia; dusko.lainscek@ki.si (D.L.); sara.orehek@ki.si (S.O.); 4Institute of Poultry, Birds, Small Mammals and Reptiles, Veterinary Faculty, University of Ljubljana, 1000 Ljubljana, Slovenia; brigita.slavec@vf.uni-lj.si (B.S.); uros.krapez@vf.uni-lj.si (U.K.)

**Keywords:** vaccine research, golden hamster, animal models, SARS-CoV-2

## Abstract

**Simple Summary:**

In 2020, Slovenia joined the global effort to develop effective vaccines and drugs to treat COVID-19. Two vaccine candidates developed in previous studies were selected and tested in the golden hamster model using four different vaccination protocols. We followed the required 3Rs principle when performing the procedures on the animals: we *refined* animal housing, handling, and measurements, including the introduction of pilot animal infection tests, and we *reduced* the total number of animals used primarily through the *replacement* procedure. Replacement was conducted by using a virus neutralisation test on cell cultures prior to infecting and killing the animals. We determined that the antibodies produced by the tested vaccines did not have sufficient neutralising properties, and the project was terminated. Approximately half of the golden hamsters that were no longer needed in the procedures were rehomed and we received very encouraging feedback from adopters.

**Abstract:**

Effective vaccines are needed to fight the COVID-19 pandemic. Forty golden hamsters were inoculated with two promising vaccine candidates and eighteen animals were used in pilot trials with viral challenge. ELISA assays were performed to determine endpoint serum titres for specific antibodies and virus neutralisation tests were used to evaluate the efficacy of antibodies. All tests with serum from vaccinated hamsters were negative even after booster vaccinations and changes in vaccination protocol. We concluded that antibodies did not have sufficient neutralising properties. Refinements were observed at all steps, and the in vitro method (virus neutralisation test) presented a replacement measure and ultimately lead to a reduction in the total number of animals used in the project. The institutional animal welfare officer and institutional designated veterinarian approved the reuse or rehoming of the surplus animals. Simple socialization procedures were performed and ultimately 19 animals were rehomed, and feedback was collected. Recently, FELASA published recommendations for rehoming of animals used for scientific and educational purposes, with species-specific guidelines, including mice, rats, and rabbits. Based on our positive experience and feedback from adopters, we concluded that the rehoming of rodents, including hamsters, is not only possible, but highly recommended.

## 1. Introduction

In 2020, researchers in Slovenia joined a global race to find effective therapies and vaccines against the COVID-19 pandemic. In this context, small animal models are essential, as preclinical studies in animals are crucial for basic and applied research in most infection studies. Several SARS-CoV-2 animal models based on ACE2 receptor homology have been studied, including nonhuman primates, transgenic mice, ferrets, cats, and hamsters [1]. Among wild-type rodents, only hamsters—*Cricetinae*—show this homology, and project proposals using hamsters have nearly tripled in Germany, for example, compared with pre-pandemic years [2]. The golden hamster, also known as the Syrian hamster (*Mesocriccetus auratus*), is the most common in publications, although it has been shown that the Chinese hamster (*Cricetulus griseus*) actually develops more severe signs of disease [3], infection in the Roborovski dwarf hamster (*Phodopus roborovskii*) can sometimes be fatal [4], and both species seem to mimic human pathology better. However, the golden hamster is well studied and readily available from laboratory animal breeders, having been recognized as a good model for many emerging infectious diseases [5], served as a model for SARS-CoV infection as early as 2005, and is essential for some specific studies [6]. They are also inexpensive, easy to handle and keep in captivity.

Vaccine efficacy (and safety) testing has been characterized by extensive use of laboratory animals. Working groups have been formed and recommendations made to develop the required 3R—Replacement, Reduction, and Refinement—methods that include a nonclinical endpoint, ultimately reducing the number of animals and decreasing severity levels of procedures on animals (refinement) [7]. Determination of antibody efficacy involves multiple testing platforms, including ELISA, lateral flow immunoassay, microsphere immunoassay, and pseudovirus systems. These assays measure the binding of antibodies to the SARS-CoV-2 spike, RBD, pre-fusion, and N proteins. However, because not all binding antibodies can block viral infection, these platforms do not measure antibody inhibition of SARS-CoV-2 infection. One possible in vitro replacement is a virus neutralisation test (VNT)—a highly sensitive and specific serological assay for detecting the presence and amount of functional antibodies that prevent viral infectivity [8]. However, for initial efficacy tests, it is not yet possible to replace the challenge procedure. Challenge of animals is used as a closed system to examine the impact of the virus and the interaction of vaccines or drugs in a living model. At least improvements—including re-evaluation of humane endpoints and overall animal welfare—must be considered. One of these aspects is the final fate of the animals. According to Articles 17 and 19 of Directive 2010/63/EU, the most appropriate decision should be taken as regards the future of the animal on the basis of animal welfare and potential risks to the environment. It is up to Member States to state under which conditions a reuse, setting free, or rehoming of the animals used in scientific procedures or bred for scientific purposes may be allowed. In recent years, there has been a particular focus on the rehoming of animals, especially appealing species such as dogs, cats, horses, and camelids, but reports of the successful rehoming of poultry, rabbits, and rodents have also been presented on institutional websites and congresses [9]. The FELASA Workgroup recently issued related guidelines [10], and designated surveys shed some light on practice in Europe and the UK [11,12].

In the present study, we tested two promising vaccine candidates based on previous research [13] on an appropriate animal model—the golden hamster—testing different vaccination protocols on 40 animals. We describe measures for refinement of all procedures on animals and show examples of replacement of the in vivo method. VNT was introduced to decide which of the vaccination protocols were promising, and with negative results (no neutralizing antibodies detected), the in vitro test actually replaced the viral challenge to test antibody efficiency. We also introduced a rehoming procedure at the end of the study.

## 2. Materials and Methods

### 2.1. Animals

Golden hamsters (*Mesocricetus auratus)* of strain HsdHan AURA were supplied by Envigo (Udine, Italy) from a UK-based breeding stock (40 animals: 20 males, 20 females) and 18 animals (8 males, 10 females) were supplied by Janvier Laboratories (Le Genest-Saint-Isle, France) of strain RjHan: AURA. Both outbred lines originate from Zentralinstitut fur Versuchstiere, Hannover, Germany. The animals were 7–8 weeks old at purchase and housed in same-sex pairs in conventional polypropylene cages with filter covers (Techniplast Eurostandard type III). Between week 8 and 10, all females and some males started to fight despite increased enrichment. Between the 10th and 12th week, we separated all animals to prevent fighting and they remained in single housing until the end of the vaccination project. A total of 40 were used for vaccine testing: 32 with the first vaccine candidate and 8 animals with another vaccine candidate. The animals were kept in dedicated rodent rooms at the Faculty of Veterinary Medicine, University of Ljubljana, which provided a controlled environment with a relative humidity of 45–60%, a temperature between 21 and 23 °C, and a 12:12 light—dark cycle. Feed and water were given ad libitum; pelleted irradiated feed contained 21% crude protein (Sniff diets S8189-S098). Irradiated wood fibre bedding was used (Lignocel, Rosenberg, Germany); for enrichment, sterilized paper strip nesting material (Sizzelenest, Datesand, Manchester, UK), sterilized aspen wood gnawing blocks (Datesand, Manschester, UK), and cardboard or polypropylene tubes for hiding were offered. 

For viral challenge tests, a pilot trial was planned and performed on 18 animals in 3 separate trials in a high-containment laboratory (Biosafety level 3) using an animal biocontainment system (Techniplast IsoRat900 N with Teklad Isoplast bedding)—designated A-BSL3 (animal biosafety level 3). For the second and third trials, animals were 8 weeks old and kept in pairs or threes as no territorial aggression occurred yet. In total, 58 animals were used for the project. 

All animal procedures were performed in accordance with the EU Directive (2010/63/EU), approved by the Administration of the Republic of Slovenia for Food Safety, Veterinary and Plant Protection (Ministry of Agriculture, Forestry and Foods) and its Ethical Committee (Decisions U34401-18/2020/8 and U34401-11/2021/9) and reports were prepared according to PREPARE and ARRIVE guidelines.

### 2.2. Vaccination

Two different routes of administration and two concentrations of the best vaccine candidate identified from previous in vitro and mouse immune response tests [13] were selected, namely, the plasmid DNA RBD-bann. A total of 32 animals of both sexes, 4 months old, were immunized and divided into 4 groups (4 males and 4 females each), which were administered either 20 or 50 µg of the vaccine plasmid DNA intranasally (i.n.) or intramuscularly (i.m.). For intramuscular administration, 50 µL was injected into the lateral thigh (biceps muscle); for intranasal administration, 50 µL was slowly pipetted into both nostrils (2–3 drops per nostril of plasmid DNA in 0.9% saline). All vaccinations were repeated after 2 weeks and specific antibody titres in serum were determined by ELISA and neutralisation properties by VNT assay. Twelve animals with the highest titres received a second booster dose of the same product i.m. after 4 weeks. The other 20 animals that were vaccinated with a lower dose and/or the i.n. route had lower antibody titres and were vaccinated 4 weeks after the second dose with two additional booster doses 3 weeks apart: 10 animals with an increased dose of naked plasmid DNA (40 µg per animal i.m.) and 10 animals with the second-best candidate from previous research: RBD-bann recombinant protein (100 µg/animal in 50 µL i.m.) coupled with squalene adjuvant (2:1 ratio) AddaVaxTM (Invivogen; vac-adx-10, San Diego, CA, USA) in order to boost the antibody production.

For the final vaccination protocol, 8 naive animals (7 months old) were used and vaccinated with recombinant RBD protein and Complete Freund’s adjuvant (Calbiochem, Merck) in a 1:1 ratio, 50 μL subcutaneously (s.c.) in two skin folds—just caudal to the elbow joint (retroaxillary) and cranial to the stifle (regio plicae lateralis).

Blood samples were collected from the animals immediately before each vaccination and 2 and 4 weeks after the last booster vaccination, always under general anaesthesia (procedure described in Refinements down the page). Blood was transferred to a mini-tube containing a gel clot activator (Microvette^®^ 500 Serum Gel, Sarstedt, Germany), allowed to coagulate for 8 h or overnight in the refrigerator, and centrifuged at 9000× *g* for 10 min. The separated serum was frozen at −22 °C until analysis.

### 2.3. Refinements in Animal Work

After transport, the animals were allowed to acclimate to the new environment for at least 1 week. If they were transferred to a BSL3 facility, they remained undisturbed for another 3–4 days. Hamsters were provided with environmental enrichments and handled by hand cupping or tunnels (especially when changing cages). Half of the food pellets were always offered on the floor of the cage to allow for hamster-specific storage behaviour. Blood samples were collected under general anaesthesia at the junction of the cranial vena cava, external jugular vein, and subclavian vein. Isoflurane was used as 4.5% in the induction chamber, maintained by 2.5–3.5% via the face mask: the animal was placed in dorsal recumbency, the puncture site was located on the left or right craniolateral to the manubrium sterni, and the needle was inserted in a caudal direction toward the contralateral hip joint [14]. The puncture site was disinfected and moistened with Spitaderm (Ecolab, Monheim am Rhein, Germany). An amount of 200–400 µL of blood per animal was collected, gentle pressure with a cotton swab for 30 s was applied on the puncture site, the gas supply was switched to an oxygen mixture, and respiration and cardiac activity were observed for a few minutes. After recovery from anaesthesia, animals were examined periodically (30 min, 2 h, 6 h, and daily thereafter) for obvious hematoma, discomfort, or pain. Initial administrations of the vaccine (intranasal—i.n. and intramuscular—i.m.) were performed under the same anaesthesia: for intramuscular administration, in the lateral position and with a face mask; for intranasal administration, the mask was removed, the hamster was grasped around the chest with the head tilted upwards, and 3–5 small drops were pipetted onto the nose, which were then inhaled. If the animal began to awaken, it was placed back under the mask (isoflurane 3%) and the remaining amount of vaccine was pipetted after the animal had lost reflexes. For booster s.c. application, only manual restraint was used, as no intermediate blood sampling was performed, thus no anaesthesia was needed and no local reaction was seen after the first dose. For both i.m. injections and blood sampling, 0.5 mL insulin syringes with fixed 30 G needles were used (BD Micro Fine Plus). For subcutaneous injection of protein suspensions and Freund adjuvant, a 1 mL insulin syringe with a 27 G needle was used (BD Microlance 3).

Randomization was performed during acclimation of the animals—one animal caretaker randomly assigned animals to cages in pairs; when pairs were separated, cages and animals were renumbered by the second caretaker. Ear notches for individual ID were made under general anaesthesia at the first blood draw. The allocation of the first vaccination protocol was conducted by the researcher, who did not know the allocation of each animal ID, except for the sex of the animals. The remaining vaccinations were allocated based on preliminary results. The researchers who performed titration, ELISA, PCR, VNT tests, or histology were blinded during the analysis.

### 2.4. ELISA Test 

ELISA was performed to determine endpoint titres for designated specific antibodies as described before [13]. Sera from hamsters after the first and second vaccinations were compared, and sera from the same hamsters before vaccination were used as negative control. Details are given in the Appendix A.

### 2.5. Pseudovirus Neutralisation System

To test the general neutralisation properties of the antibodies detected in hamster sera, a pseudovirus (PV) neutralisation assay was carried out as described before [13]. Details are given in the Appendix A.

### 2.6. Detection of Neutralising Properties of SARS-CoV-2 Antibodies

Serum samples were tested for the presence of neutralising SARS-CoV-2 antibodies using a virus neutralisation test (VNT). The SARS-CoV-2 strain Slovenia/SI-4265/20 was provided by the European Virus Archive and the Vero E6 cell line (African green monkey kidney cells) was provided by ATCC: VERO C1008; CRL-1586 [15]. The positive reference serum (EURM-018 human serum, JRC, EC) was used as a positive control. More details are given in Appendix A. 

### 2.7. Pilot Trial for Viral Challenge Test

In the first pilot trial, 6 naïve hamsters (2 males, 4 females) were intranasally inoculated with 25 μL of the SARS-CoV-2 virus strain Slovenia/SI-4265/20 (European Virus Archive) with a titre of 5 × 10^4^ TCID 50/mL diluted in Eagle’s Minimum Essential Medium (EMEM), observed daily for clinical signs, and killed with CO_2_ and exsanguination on days 2 (2 female), 4 (1 male, 1 female), and 7 (1 male, 1 female) after inoculation to determine the best time point and tissue samples. Nasal conchae, trachea, lung tissue, duodenum, and whole brain were collected. FLOQSwabs (Copan, Italia) from the caudal nasal cavity, trachea, lung’s cutting surface, and duodenum were also taken for RT-qPCR. Based on the initial results, a second trial with a group of 6 animals (3 males and 3 females) was conducted. The animals were inoculated intranasally (i.n.) with 50 μL (25 μL per nostril) of the virus with a titre of 5 × 10^4^ TCID_50_/mL, monitored daily for clinical signs, and killed with exsanguination (cardiocentesis) under inhalation anaesthesia on the fourth day after inoculation. Because clinical and pathologic signs were low, we increased the viral load and tested on an additional 6 animals (3 female, 3 male)—inoculated i.n. with 50 μL (25 μL per nostril) of the virus with a titre of 1 × 10^6^ TCID_50_/mL—designated as the third pilot trial. Animals were monitored and weighed every 24 h and killed by exsanguination under inhalation anaesthesia on day 4. Blood and lung tissue samples were collected—the entire left lung lobe was placed in 4% buffered paraformaldehyde for histological processing, and the entire right lung was weighed and homogenized in sterile Eagle’s Minimum Essential Medium (EMEM). The clinical score sheet was used for signs of lethargy, ruffled coat, hunched posture, laboured breathing, nasal discharge, cyanosis, and facial grimace: 1 for mild and 2 for severe manifestation (daily score above 6 was considered as humane endpoint; none of the animals reached this). The same trained person observed the animals and filled in the evaluation forms. She was not blind to the study, but she was blind to animal ID each time to start with as the cage position might have changed at the first morning inspection by the animal caretaker. Animals in A-BSL3 were identified by cage cards and abdominal marking (blue pen) to separate between 2 or 3 animals in the cage only upon close inspection.

### 2.8. Determination of Virus Titres

Virus titres were determined by titration of sample homogenate suspensions on Vero E6 cells and measured as the 50% infectious tissue culture dose (TCID_50_/mL) [16,17,18]. Some details are available in the Appendix A.

### 2.9. RNA Extractions and Quantitative RT-PCR

The swabs of organs were individually vortexed in 2 mL phosphate-buffered saline for 2 min prior to genomic nucleic acid extraction. Total RNA and DNA were extracted from 140 µL of sample supernatant by the QIAamp Viral RNA Mini Kit (Qiagen, Hilden, Germany) according to the manufacturer’s instructions. Viral RNA was detected with the real-time assay RT-qPCR targeting the E gene of SARS-CoV-2 using the primers and probe described by Corman et al. [19]. RT-qPCR was performed by QuantStudio 5 (Thermo Fisher Scientific, Waltham, MA, USA) using 2 µL of the extracted total RNA.

### 2.10. Immunohistochemistry

The left lung was fixed in 4% buffered paraformaldehyde for 3–4 days, embedded in paraffin, and cut into 5 µm horizontal sections and stained either with HE or the location of the virus with SARS-CoV-1/2 Spike Protein (2B3E5) mouse mAb (#52342, Cell Signaling Technology, Danvers, MA, USA) was labelled. More details on this procedure can be found in the Appendix A. 

### 2.11. Reuse and Rehome

The prospect of reusing or rehoming the animals was not included in the project proposal, as all animals were expected to enter the A-BSL3 facility for viral challenge and be killed there. Based on the results during the vaccination process, it was decided to omit the viral challenge procedure on vaccinated animals because antibodies produced did not have (detectable) neutralising properties and infecting the animals would not bring any new data needed for the purpose of the project. The facility animal welfare officer was consulted to approve the reuse or rehoming of the 40 surplus animals—the animals that were vaccinated but not exposed to the virus. The institutional designated veterinarian certified that most of the animals were in good health and posed no threat to the environment or human health, so they could be reused or rehomed, even though they were about halfway through their life expectancy at that point: 16 months old and life expectancy is between 2 and 3 years. 

A total of 16 animals in suboptimal condition were sacrificed for tissue samples for basic research and controls. In the meantime, simple socialization measures were implemented, primarily through giving fresh food treats, frequent handling, and alternative environmental enrichment. Another 5 animals that did not adapt well to handling were deemed unsuitable for rehoming as pets. Adopters were initially institutional and later external. Prospective adopters contacted the person responsible for the hamsters, an interview was conducted to confirm the adopter’s knowledge of hamster care, a simple adoption contract was signed, and voluntary follow-up meetings were held over the following months. In the adoption contract, there was also basic information on procedures the animals were subjected to and approvals of the designated veterinarian and welfare officer. DNA vaccines to date do not persist or even biodistribute throughout the body of the vaccine recipient when delivered parenterally into muscle, subcutaneous tissue, or dermal layers. The local response to plasmid DNA inoculation is that cells take up the plasmid and then express the immunogen(s)-encoded mechanisms; the nucleic acid is degraded by normal molecular mechanisms. As a consequence, the plasmid DNA clears from the injection site over time [20]. A minimum of 2 months passed between the last vaccination and adoption and adopters expressed no concerns on vaccination methods used.

Out of 40 animals, 19 animals were designated fit for rehoming and all found new homes. Eighteen months after the last adoption, a simple anonymous online survey was conducted via a local popular webpage, www.enka.si. The link to the questionnaire was sent to 17 adopters, and 11 responded. The (translated) questionnaire is added in the Appendix A.

## 3. Results

Sixty hamsters were planned to test the plasmid DNA RBD-bann vaccine (in low and high doses and in two modes of administration, i.m. and i.n.) as well as the protein-based RBD-bann. The supplier was only able to supply forty hamsters, which we divided into four groups for the plasmid DNA vaccine and one group for the protein-based vaccine, with each group consisting of four male and four female hamsters. Prior to vaccination, the hamsters tested negative for SARS-CoV-2 (oropharyngeal swab to rule out possible exposure to the virus in the environment) and a blood sample was taken. This was used as a control—all antibody titres were compared with a control serum from each individual. Protein RBD bann was tested on 8 naïve hamsters after the initial ELISA and VNT results and also on 10 animals in the form of booster doses after two doses of the DNA vaccine. The diagram in Figure 1 explains the workflow.

With 18 animals from the new supplier, we conducted pilot tests with the virus in an animal biosafety level 3 (A-BSL3) laboratory. None of the vaccinated animals entered A-BSL3 and were not infected with the virus.

### 3.1. Serum Antibodies 

As expected, only the booster dose resulted in high levels of IgG titres in the hamster serum presented in the graph in Appendix A. At a lower dose of the tested vaccine (20 µg per animal), some animals showed an insufficient immune response, which was more evident with the plasmid DNA product. 

### 3.2. Pseudovirus Neutralisation Test

Neutralisation of the pseudovirus showed little effect, and insignificant lower or higher luciferase signal levels did not correspond to group assignment (Appendix A).

### 3.3. Virus Neutralisation Test 

No neutralising antibodies were detected by VNT in any of the hamster serum samples tested; all dilutions resulted in cytopathogenic effect (CPE—Appendix A). The positive control serum VNT titre was 1:32. 

### 3.4. Refinement of Viral Challenge Procedure—Pilot Control Groups

To determine the best time for euthanasia, tissue, and method of sampling, pilot experiments were conducted with a minimal number of animals. Non-vaccinated animals were infected with 25 µL 5 × 10^4^ TCID_50_/mL SARS-CoV-2 strain Slovenia/SI-4265/20, day 4 after inoculation was selected as the best time for the clinical endpoint, and swabs from the caudal nasal cavity were more representative than swabs from the oropharynx but were time-consuming to obtain in an A-BSL3 facility. In the duodenum and trachea, viral RNA was either not detected or was very low (Ct values above 33—see Appendix A). In the second trial, we doubled the amount of viral particles by applying 50 µL (25 µL per nostril) of 5 × 10^4^ TCID_50_/mL to 6 unvaccinated hamsters and all were euthanized on day 4. RT-qPCR of viral RNA and viral titres on Vero E6 cells roughly correlated, but levels were low and clinical signs were minor; in addition, contrary to expectations, females gained weight (Table 1). By increasing the viral load 100× (50 µL of undiluted stock—5 × 10^6^ TCID_50_/mL) in the third trial, we observed more obvious clinical signs: lethargy and ruffled coat, and 50% of animals had a mild nasal discharge and showed laboured breathing. The amount of virus in the lung tissue detected by viral titre from lung homogenate increased, viral RNA RT-qPCR Ct values dropped, and the ΔCt between the second and third trial were significantly different at *p* = 0.002 as presented in the Appendix A. However, the increased viral load could not be confirmed with higher immunohistochemistry scoreof spike protein antigen (Appendix A and scoring in Table 1).

The use of 6 extra animals (third trial) or pilot trials as a whole (18 animals) might contradict the reduction principle of 3R but was considered as refinement as the virulence of specific SARS-CoV-2 strains for golden hamsters was not known. None of the vaccinated animals were subjected to the viral challenge procedure.

### 3.5. Rehoming

Of the 40 animals vaccinated, 1 developed otitis media, was treated at the Faculty’s Clinic for birds, small mammals, and reptiles, and was eventually adopted by the attending veterinarian with the approval of the institutional animal welfare officer. The designated veterinarian performed the clinical examination of the remaining animals. Three were in worse condition, likely due to their advanced age (over 1 year old), and were euthanized. Another 12 animals were selected for reuse for scientific or educational purposes, humanely killed, and tissue samples collected. Socialization measures were taken for the remaining 24 animals. Five of the animals did not respond well to frequent handling, usually hiding in the tunnel when staff were present, so they were not candidates for rehoming as pets. A total of 19 hamsters were eventually adopted; the first six adopters were internal, the next seven semi-internal (veterinary students and trainees in the LAS course of October 2021), and the rest external. The word was spread through friends and relatives. Internal adopters provided frequent feedback, and a year and a half later, a general opinion was obtained through an anonymous online survey (translation available in the Appendix A). Of 17 invitations sent (2 contacts were lost), 11 responded; the main results are shown in Figure 2. All feedback received was positive and adopters would recommend continuing the practice. A slight dissatisfaction of four respondents (they would not recommend the next adoption and were not completely satisfied with the adoption procedure) was related to the result that their four hamsters had a shorter life than they expected and had some health problems, e.g., the appearance of tumours in one. For confidentiality reasons, no further analysis was carried out. The translation of the questionary with consent request can be found in Appendix A.

## 4. Discussion

Effective vaccines are needed to combat the emerging viral epidemics. In this study, we evaluated different deliveries of promising vaccine candidates against SARS-CoV-2 in the Syrian hamster model. Unfortunately, the goal of our application-oriented research project was not achieved—the vaccine candidates elicited an immunological response, but we could not confirm the neutralising properties of the antibodies produced in the golden hamster. In 2021, almost every country was trying to develop a vaccine for SARS-CoV-2, the vast majority focusing on spike protein as the most immunogenic structural protein of the virus. The success of a vaccine to fight a pandemic is dependent on durability and stability of neutralization antibody titres [21]. Mutations in antibody epitopes on the spike protein can result in increased viral resistance to neutralizing antibodies and have been associated with reduced vaccine effectiveness with recent variants like omicron. Although variant-specific vaccines and novel monoclonal antibodies are being developed for optimal activity, booster immunizations with the old products are also investigated as they can significantly increase serum neutralizing activity [22]. Humoral immune response—the antibodies—is the best-defined correlate of protection against the SARS-CoV-2 infection; however, nowadays with emerging variants, more focus is put also on cellular components: specific B cells are important for long-term protection from the disease and T cells specific for SARS-CoV-2 can protect from severe disease [23]. At the end of 2020, however, the most important goal was to slow down the spread of the disease, and only efficient neutralisation of the virus was the aim of the project funded by the Slovenian Research and Innovation Agency and the Ministry of Defence. We have no explanation for our unexpected result—no neutralising properties of the antibodies—and no further studies on the causes and other types of immune response were conducted with additional animals, as this would create an unauthorised deviation from the official decision of the Slovenian Ministry of Agriculture, Forestry and Food and its Ethics Committee. However, we did emphasize the importance of the 3R principle in practice. The alternative procedure reduced the total number of animals in the study: the sero-neutralisation test or VNT turned out as a complete replacement for the efficacy test with viral challenge of the animals as no new data would be obtained for the scope of the project. Because VNT continued to show a negative result despite booster vaccinations and changes in the vaccination protocol, and was backed up with a negative PV test, we decided to discontinue the trials. None of the vaccinated animals underwent viral challenge (moderate procedure according to classification of the severity of procedures on animals—Directive 2010/63/EU) and they were not killed for this purpose, meaning the reduction in severity was achieved. Also, the total expected number of hamsters in the project (120) was reduced because none of the vaccination protocols were repeated (i.e., larger groups for better statistical evaluation and control group with placebo injections), no alternative modes of administration were tried (oral patches and enhanced intranasal application), and no product safety testing was required. 

In the meantime, while the vaccination protocol was ongoing, we subjected a small group of animals to a pilot viral challenge test in order to establish an appropriate protocol for the planned tests of vaccine efficacy. Pilot studies are recommended to design a study with a needed sample size estimation when the general results of infection procedures are unknown [24] and when the laboratory/institution is confronted with new methods, equipment, animal species, etc. [25]. In our study, a pilot trial was planned to test the workflow in a newly equipped A-BSL3 facility and to determine endpoint measurements; 12 animals in two consecutive trials were initially used for this purpose. We followed the protocol descriptions from the publications on the use of Syrian hamsters at that time and infected the animals with similar viral loads between 1 × 10^4^ and 1 × 10^5^ TCID_50_/mL [26,27]. Because the observed clinical signs and virus isolates from lung tissue were very low, we tested a higher dose in six additional animals with better results (higher clinical score and lung viral load). This dose would be used in further procedures to achieve robust infection in control animals, because we want to avoid difficulties in distinguishing between mild infection and possibly only mild improvement. Pilot trials seem to contradict the reduction principle if all goes well. But if animals with a promising vaccination response (and control groups) are infected with suboptimal viral load, the efficiency parameters will likely be insignificant, requiring the whole protocol to be repeated, thus using even more animals. Higher doses have been used by some other investigators [28], and even the oropharyngeal route could more effective [29]. Alternatively, a more infectious viral strain or a completely different model animal could be considered. The results of our pilot tests confirmed the need to consider this as a refinement method when planning research projects including animal procedures. 

The mild clinical signs observed, including very little decrease in body mass (or even increase in females), must not only be due to the (initially) low inoculum dose [30] but are also likely strain-dependent. Increase in body weight has been noticed when animals were infected with a lower dose, but the lung pathology was comparable to higher doses, where animals lost weight [31]. The local specific strain Slovenia/SI-4265/20 has not been tested elsewhere and we have not used any other strain for infecting the animals for comparison. Although this is one of the early isolates in 2020, its virulence for golden hamsters could be low, as has been shown for some later variants [32,33]. Body mass is a robust measurable parameter; on the other hand, observation of respiratory patterns and general well-being is highly subjective, and standardized clinical scoring for golden hamsters or similar models is difficult to develop [30]. For the best refinement solution, automated home cage monitoring systems would be a perfect addition, especially because hamsters are nocturnal and the prominent changes in their behaviour can easily go unnoticed.

The immunohistochemistry score in our pilot control group was weak and only slightly improved after increasing the inoculation dose. Most of the positive signals were located in the bronchi and only sparse interstitial locations in two individual animals. It should be noted that pathologic changes are not evenly distributed and, therefore, a wide transverse section of the entire lung lobe, ideally the left non-lobulated one, is needed [34]. At least five semi-serial sections are required for reliable quantification with immunohistochemistry, which is slightly more time-consuming than histopathology scoring, where a grid scoring was proposed on one slide [31]. For optimal pathologic evaluation, euthanasia and sampling should be rather conducted on day 5 or 6 post-inoculation, as the virus enters/replicates interstitially after day 4 [35] and exact pathological scoring was not used at this stage in our experiment. For appropriate refinement, decision making for the clinical endpoint should be discussed—for optimal histopathology and clinical score results, hamsters should be left in (moderate) distress for longer (4–7 days). For therapies and vaccines targeting viral replication, sample analysis (virology) at the peak of viral replication between 2 and 3 dpi may be more useful.

Researchers always strive to minimize the number of animals used as long as a relevant statistical analysis can be performed. When working with dangerous infectious agents, downsizing animal numbers may even be enforced due to the limited capacity of A-BSL3 facilities. The risk of over-reduction exists and thus challenges the reliability and reproducibility of the studies—the factors that are often referred to “Rs beyond the 3R”. For our study, we managed to organise a capacity to house up to 60 hamsters simultaneously in a renovated A-BSL3 but were unexpectedly faced with limited availability of the species. Instead of the planned 60 hamsters for different vaccination protocols and three control groups with placebo injections, we could initially acquire only 40 animals. A decision was made to optimise the experiment design: test all the planned protocols, but for the controls/blanks, use the sera of the same hamsters before the vaccination. Such a reduction principle—comparing responses at later time points to a baseline level—is commonly used in toxicology [36]. We did plan to retest only one most promising protocol, including more animals depending on variability and adjusting the dose or adjuvant if needed as well simultaneously using the placebo group, e.g., DNA vector without the insert. As none of the vaccination regimens even came close to the wanted result, the repeats and further steps (i.e., safety) were not conducted, and this ultimately reduced the number of animals. 

Regarding the reliability of the studies, we need to highlight the sex differences, as we have observed more pronounced clinical symptoms in males and this has subsequently been well documented [37,38]. Studies examining only one sex (either males only or females only with no clear explanation of selection) may therefore overestimate or underestimate vaccine efficacy, dangers of new variants, or transmission through domestic animals [27,29,39]. Studying both sexes has been a guiding principle in biomedical research for so long that the decision to use only one sex can hardly be justified [40,41], even if it seems to contradict the reduction principle. Furthermore, for real vaccine or drug efficacy, the number of animals used should likely increase by including different ages of animals, as disease progression is different in aged animals [42]. We started with vaccinations when animals were 4 months old, and eight in the last protocol tried were 7 months old. Most publications have used very young hamsters, probably to speed up research when faced with pandemics. We can argue this is a drawback; using prepubescent animals for this kind of translational study is not exactly beneficial and rather animals in their prime adult age should be considered. Therefore, also the wanted (starting) age requires more careful planning in the inclusion of model animals in the future. 

We found no reports of negative results that would explain the lack of neutralising properties of hamster antibodies, and overall, not that many papers describing vaccine testing in golden hamsters were found. It is likely that most laboratories started using genetically modified mice (hACE2 mice and further variants) when they became more readily available, because after the initial boom, the number of studies using hamsters declined slightly [2]. However, we can surmise that more researchers encountered obstacles in using this animal model but did not report it. The VNT test should not yield false-negative results because it has not been shown to be species-dependent [43]. Withholding negative results from publication—known as publication bias—can seriously distort the literature and consume scarce resources through lengthy and perhaps even futile research. The research community is ethically obligated to make the best use of the results of animal studies, which is not the case when negative results are not published [44]. To overcome this publication bias, initiatives to pre-register animal studies have been proposed [45]. To date, there are only two reliable animal registries: Preclinicaltrials.eu and the Animal Study Registry (animalstudyregistry.org). Unfortunately, neither of them show any results with the keyword “hamster” (last accessed 15 June 2023). Also, while vaccine efficacy in the first weeks after vaccination can be readily studied in animal models, monitoring the duration of vaccine-induced immunity remains a challenge due to differences in metabolism and life expectancy between rodents and humans [35], and the more data published, the better.

Refinements such as environmental enrichments, friendly handling, and general anaesthesia for all inoculations and blood sampling have been introduced for the general welfare of the animals, even though minimal pain procedures were unlikely to affect the final immunological outcome. We hypothesize that minimizing the pain and other negative associations with humans contributed to rapid and smooth socialization in most animals, as confirmed by adopter feedback. It is important to note that this species exhibits burrowing behaviour [46], and they all used the “Sizzlenest” paper strips to build nests but left cotton fibre “Nestlet” mostly intact. Also, most hamsters used the bent (at right angle) plastic tunnels to seek refuge when startled but rarely slept in them. Some hamsters preferred to nibble on aspen blocks and others the cardboard tunnels; therefore, the choice should be given. An important aspect of hamster welfare is to check that they can reach food through the grates as their snouts are shorter and wider than those of rats or mice. It is best to offer them the food inside the cage [47]. If the food is scattered in the bedding, this encourages the hamsters’ natural behaviour of gathering and storing their food in a specific location. They regularly urinate in one corner of the cage only, so the food is not spoiled by urine, and this also enables partial cage cleaning. In the wild, the golden hamster is a solitary animal, but juveniles are supplied by breeders conditioned to group housing, and housing with littermates may actually be recommended, at least for males [48]. In our experience, it was possible to keep hamsters in same-sex pairs or groups of three for short-term experiments, such as virus challenge trials in A-BSL3, where acclimatization lasted a week, the experiment lasted a maximum of another week, and the cage floor surface exceed 1800 cm^2^. With vaccination protocols, animals were kept longer in the conventional facility, and soon after reaching sexual maturity, all females began to fight, inflicting bite wounds. To create comparable conditions, we separated both sexes and switched to single housing in these procedures.

The final refinement represented the decision to rehome some of the animals. The rehoming process was initially met with some scepticism by the research team but was then supported by the animal welfare officer, approved by the designated veterinarian, and accepted with great satisfaction in the end. Most adopters were overjoyed to give the animal a second chance in life and would gladly adopt a similar animal next time. We do not know the opinion of the adopters who did not respond to the survey (35%)—their experience may be different but would not be the deciding factor in this case. To date, there are no specific institutional (or national) guidelines or protocols regarding rehoming animals used or bred for research purposes. In Slovenia up until now, only farm and wild animals that have undergone non-invasive or minimal harm procedures have been considered suitable for “return to herd” or “set free” in the case of wildlife. Once the ice was broken, future project proposals may consider rehoming laboratory animals at this and other institutions as public interest—including for rodents—appears to be greater than scientists would generally hope for. Success stories, including rehoming rodents, come from other European countries. Some have chosen to work with animal welfare NGOs (animal protection and animal rights organizations) [9]. This is probably the best solution as long as the NGO staff can establish a good relationship of trust and communication and work hand in hand with the scientists and regulators to ensure that they are sending an appropriate message to the public.

As laboratory animal science faces significant challenges, openness and transparency with the public is now encouraged. Citing MacArthur Clark [49], “A balance is demanded between the needs of the science and the needs of the animals since this balance supports and sustains the public’s confidence in ethical review processes and the regulatory oversight of the use of animals in science. Tinkering with this balance, either excessively in favour of science or excessively in favour of animal welfare, will lead to loss of public confidence and erosion of the conditional permission society grants to enable animal research to continue.” We believe that a process of rehoming surplus or retired laboratory animals with success stories published in the dedicated media can be a step forward in confirming this balance.

## 5. Conclusions

Refinement is a ubiquitous tool in almost every step of research projects involving procedures on animals. Pilot trials are highly recommended when new techniques, equipment, infectious agents, etc., are used because it confirms the observation of the reduction principle. In our pilot viral challenge test, we found that we needed to use a higher viral load of a particular virus strain to elicit a sufficient clinical response as originally planned—and this would prevent the need to repeat the entire experiment with vaccinated animals. The virus neutralisation test is an example of an in vitro prerequisite that may reduce the severity or even total number of animals in the project. Because it could not demonstrate the neutralising properties of the hamster antibodies on vaccine candidates tested in our project, it served as a replacement for the in vivo viral challenge. The surviving golden hamsters were successfully rehomed in our study. Positive attitudes among both scientists and adopters of retired or surplus laboratory animals can improve public attitudes toward the use of animals for scientific purposes and help restore general confidence in science.

## Figures and Tables

**Figure 1 animals-13-02616-f001:**
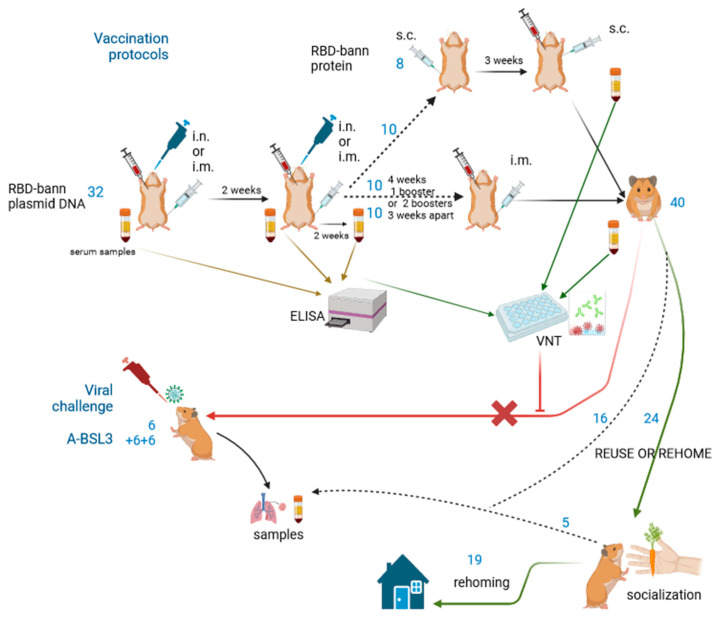
Diagram of workflow for vaccination, sampling, and pilot viral challenges. The blue numbers indicate the number of animals used in each run, initial vaccinations were half i.m. (syringe symbol) and half i.n. (pipette). ELISA testing of serum was crucial for further steps in the vaccination protocols: 10 animals with highest IgG titres received a second booster of 50 µg plasmid DNA, 10 animals received 2 boosters (3 weeks apart) with a higher dose of plasmid DNA than that initially (40 µg instead of 20 µg) and 10 animals received 2 boosters of protein RBD-bann 3 weeks apart, all i.m. An extra 8 animals were vaccinated with protein RBD and complete Freund adjuvant (top line). The virus neutralisation test (VNT) served as the eventual replacement method for viral challenge. There were no neutralization properties, the CPE effect was seen in all dilutions, and the continuation of procedures to viral challenge was stopped (red line). Health checks were performed on 40 surviving hamsters and a decision was made as to whether they should be reused or rehomed. A total of 24 hamsters underwent a simple socialization process and 19 eventually found new homes. Viral challenge (i.n. inoculation, red pipette) was tested in pilot trials on non-vaccinated animals. Diagram drawn in BioRender.

**Figure 2 animals-13-02616-f002:**
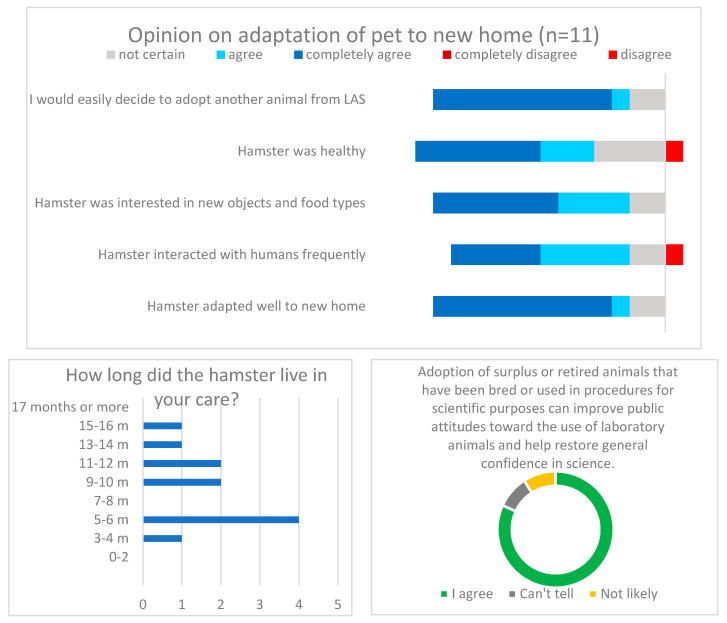
The main findings from the survey among adopters 1.5 years after the rehoming procedure. None of the responders choose the option of “completely disagree” in the first panel. No. of responses = 11 (65% success rate).

**Table 1 animals-13-02616-t001:** Second and third trials for pilot viral challenge control group with intranasal inoculation of 50 µL of 5 × 10^4^ or 5 × 10^6^ TCID_50_/mL SARS-CoV-2 strain Slovenia/SI-4265/20 to achieve evident clinical scoring and tissue viral load. Differences between male and female animals were noted especially in body mass gain/loss.

Trial	Sex	Lung Virus Titre TCID_50_/mL	Lung Histology *	Body Mass Change	Clinical Score **
Viral challenge 50 µL 5 × 10^4^ TCID_50_/ml	females (n = 3)	detected 1/3 (8.62 × 10^2^)	range − to +	average +4%	1
	males (n = 3)	detected 1/3 (1.86 × 10^4^)	range −/+ to +	average −2%	1
Viral challenge 50 µL 5 × 10^6^ TCID_50_/ml	females (n = 3)	detected 3/3 (1.56 × 10^4^–6.34 × 10^4^)	range − to ++	average +5%	2–3
	males (n = 3)	detected 3/3 (1.49 × 10^4^–1.07 × 10^5^)	all +	average −2.5%	1–3

* Semi-quantification of immunohistochemistry, colorimetric detection of spike protein in lung bronchi and interstitial tissue on 5 semi-serial sections of 1 lung lobe. Maximum score is +++. ** Clinical scores 1 point for mild and 2 for prominent: lethargy or slow arousal response, ruffled coat, abnormal posture, dyspnoea, ocular or nasal discharge, diarrhoea. Score above 6/day was considered humane endpoint.

## Data Availability

The data presented in this study are available on request from the corresponding author.

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
