# Peer review of "Rehoming and Other Refinements and Replacement in Procedures Using Golden Hamsters in SARS-CoV-2 Vaccine Research"

_animals, 2023, doi:10.3390/ani13162616_

Round 1

Reviewer 1 Report

Review:  “Rehoming as the ultimate refinement…”

Big Picture:

This is a thoughtful and well-written article and a good contribution to laboratory animal welfare.

The title is a bit misleading, as the article covers many refinements (as well as replacement and reduction strategies), plus – catchy though it sounds – I’m not sure rehoming is the ultimate refinement, given all the pain and suffering you are working to reduce.

The manuscript combines two different papers:  one, a project in vaccinology exploring hamsters’ immune response to possible vaccine candidates and the other, an evaluation of hamster adoption.  Notes on refining the experiment work well in either of these two papers.

I am not sure the details about the science itself (the analysis of the antibody levels and the activity of the antibodies, for example) is necessary for this paper focused on 3Rs of the animal use. If this description of the vaccine trials is important, perhaps you can give a bit more detail on the vaccines themselves (this may be proprietary info that you cannot divulge) so that others can see these have already been tested and found to be ineffective.

Suggestion:  remove the evaluation of the vaccines from this paper, and make it solely about adoption and other alternatives.

I had many questions in reading the methods, and while most of them are described in the Discussion, with the rationale, I would still add a few words in the Methods to address these questions

·      Line 95:  Females and males?  How many of each? 

·      Lines 100-101: state why you moved them to single caging – prevent fighting (thus, improve animal welfare)? Experimental needs?

·      L. 110: How many animals in the pilot trial?

·      L. 123: state (if true) that animals were manually restrained for the in and im vaccines.

·      L. 139:  How did you collect blood (what vein? Anesthetic?)

·      Ll. 150ff:  good description of anesthetics, but might also be good to mention this earlier, in case people try to replicate your work and look more at Methods than at Refinements.

·       

Some other small notes:

L.107 – well done on offering multiple forms of nesting material

L. 147:  very good, gentle handling

148 – offering some food on floor for caching is good

L. 166:  how did the caretakers randomize the animals? Random number table?

Lines 214ff:  Good description of endpoints. Some are subjective, so it’s relevant if the same person did all the scoring, whether they were blinded to the study, whether they had training on reading subjective read-outs like grimaces

Line 218: How many animals reached your humane endpoints?

A general comment: unless they are relevant to the 3Rs, some of the description of how you processed samples may not be relevant or necessary (ELISA, pseudovirus, immunohistochemistry, RNA assays, etc.) neutralization, .  If they ARE relevant to the 3Rs, be sure to point that out --- for example, were you good at getting multiple types of data from the same few animals (reduction) or did you do assays in vitro that some people would perform in vivo?

Line 258 “Based on the results during the vaccination process, it was decided to omit the viral challenge procedure on animals.”  BECAUSE  . . .     You needed to verify antibody levels first? Or  . . . ?

277-278  - this diagram is a little confusing

·      What do the blue pipettes signify?

·      Adding “24” to the solid green line “Reuse or rehome” to account for all 40 animals in that first branching path may clarify

L. 315:  How did you determine the dose of virus to try?  You started low, a Refinement that may have spared animals serious clinical illness, but then used more animals in the long run as you had to keep increasing the dose.

L. 354:  Just curious: how did you treat the otitis media and how had it presented?

L.415 ff :  No monitoring of body temperature, correct?

L.461:  great point about publication bias. 

Ll.499-517 great discussion on rehoming norms in Slovenia, and makes it even more impressive that they were able to convince researchers to rehome these animals

Author Response

This is a thoughtful and well-written article and a good contribution to laboratory animal welfare.The title is a bit misleading, as the article covers many refinements (as well as replacement and reduction strategies), plus – catchy though it sounds – I’m not sure rehoming is the ultimate refinement, given all the pain and suffering you are working to reduce.

 Thank you. Yes, we thought the catchy title would aquire more interest of prospective readers but we probably exagerated, so we changed it slightly.

The manuscript combines two different papers:  ………… If this description of the vaccine trials is important, perhaps you can give a bit more detail on the vaccines themselves (this may be proprietary info that you cannot divulge) so that others can see these have already been tested and found to be ineffective.  Suggestion:  remove the evaluation of the vaccines from this paper, and make it solely about adoption and other alternatives.

As suggested by the Editor we moved the results of in vitro tests to Supplement material. We believe these data is crucial to publish exactly due to ineffective outcome but would be somewhat inconclusive to publish as standalone paper (properties of antibodies or other immune response were not investigated further). In this form they support the main point of the paper - refinement procedures and are accessible to the public at the same time.

 I had many questions in reading the methods, and while most of them are described in the Discussion, with the rationale, I would still add a few words in the Methods to address these questions

  • Line 95: Females and males?  How many of each?   Added in the text
  • Lines 100-101: state why you moved them to single caging – prevent fighting (thus, improve animal welfare)? Experimental needs?

Yes, to prevent further fighting. We tried to avoud this (based on suplier claims they can be kept in smal groups provided sufficient enrichement.) 50% percent of males would likely do fine, but none of females. To have groups then equalised we separated them all. Animals that were moved to BSL3 soon after arrival were young enough and did not yet exhibit agression so for their welfare they were kept in pairs or threes.

  • L. 110: How many animals in the pilot trial? added
  • L. 123: state (if true) that animals were manually restrained for the in and im vaccines.

No, they were in anasthesia, following the blood draws.

  • L. 139: How did you collect blood (what vein? Anesthetic?) as below
  • Ll. 150ff: good description of anesthetics, but might also be good to mention this earlier, in case people try to replicate your work and look more at Methods than at Refinements. 

In order not to repeat parts of the text it is mentioned at subtitle Vaccinations to see under subtitle Refinements. Both are under Methods and some extra rearrangement was done to clarify the procedures.

Some other small notes:

L.107 – well done on offering multiple forms of nesting material. ?

  1. 147: very good, gentle handling ?

148 – offering some food on floor for caching is good ?

  1. 166: how did the caretakers randomize the animals? Random number table?

No, just spontaneous. One caretaker at the arrival of the animals, later the other caretaker at the second (splitting) in a different logistics, so no one knew the final outcome. No exclusion criteria were needed, all animals were in good health. At the first vaccination, one researcher decided where to start, animals were brought in from another room, another resercher searcher who prepared the vaccines decided which dose to start with. The animals received their final IDs (ear notch under general anaesthesia) at this time. It seemd to work perfectly.

Lines 214ff:  Good description of endpoints. Some are subjective, so it’s relevant if the same person did all the scoring, whether they were blinded to the study, whether they had training on reading subjective read-outs like grimaces

We are small facility – the only caretaker at newly equipped BSL3 was freshly trained and still worked under (my) supervision. Myself - M.S. did all the scoring and also set up many of the protocols, also educating others at the same time. I am trained in working with various animal species (DVM + many LAS trainings). I could not be blinded to the study, but I always try to inspect the animals first without observing their ID's (cage cards) and ask for second opinion the designated veterinarian. I have added a bit of clarification here.

Line 218: How many animals reached your humane endpoints?

None - clinical signs were mild. But there was procedural endpoint on day 4.

A general comment: unless they are relevant to the 3Rs, some of the description of how you processed samples may not be relevant or necessary (ELISA, pseudovirus, immunohistochemistry, RNA assays, etc.) neutralization, .  If they ARE relevant to the 3Rs, be sure to point that out --- for example, were you good at getting multiple types of data from the same few animals (reduction) or did you do assays in vitro that some people would perform in vivo?

ELISA, pseudovirus and VNT were all used as refinements in the project to choose which – if any – animals were subjected to further procedures. It was decided to move most of those results to Supplement material, so we can move the detailed description to Supplement as well (other reviewers actualy ask for a some more text or reference for better clarification).

Line 258 “Based on the results during the vaccination process, it was decided to omit the viral challenge procedure on animals.”  BECAUSE  . . .     You needed to verify antibody levels first? Or  . . . ?

Antibody levels were tested with ELISA. VNT and pseudovirus system were in vitro methods to test neutralising properties of the antibodies introduced as refinements. Based on these results we could decide which vaccination regime was good enough to test the vaccine eficacy with in vivo challenge test, likely reducing the number of animals needed to be killed. As there was no neutralizing effect at all, it made no sense to challenge and kill the animals, as only effective vaccine was the goal of the project (other immune responses were not in the interest of partners and financers). Therfore in this case VNT was actually a replacement method.

277-278  - this diagram is a little confusing

  • What do the blue pipettes signify? intranasal application (it was either i.m. or i.n. and this is added as text now)
  • Adding “24” to the solid green line “Reuse or rehome” to account for all 40 animals in that first branching path may clarify Done
  1. 315: How did you determine the dose of virus to try? You started low, a Refinement that may have spared animals serious clinical illness, but then used more animals in the long run as you had to keep increasing the dose.

Initial dose was reproduced from publications we reffered to at the time (later more papers with also higher doses were published). As the clinical signs were too mild we increased the dose twice. Indeed, more animals were used for pilot trials than intended. On the other hand, if pilot test(s) in ABSL3 was not done at all and challenge test was performed, we could be faced with the need to repeat the challenge –up to 40 animals vs 18 extra. Therefore the need to include pilot tests is discussed in the Discussion.

  1. 354: Just curious: how did you treat the otitis media and how had it presented?

This animal developed severe head tilt and discoordinated movement, spinning around longitudinal axis. It was taken to Veterinary Faculty Clinic for birds, small mammals and reptiles for diagnostics and no conclusive cause was found. To my knowledge treatment was symptomatic (at the Clinic – it was not returned to the LAS facility) and the animal fully recovered in 4 weeks. It was adopted by treating veterinarian (if more details are needed you are welcome to write to correspondence e-mail address).

L.415 ff :  No monitoring of body temperature, correct?

 True.

L.461:  great point about publication bias. 

 Yes, therefore I see it fit to include all the negative results in this paper, even as Supplement material.

Ll.499-517 great discussion on rehoming norms in Slovenia, and makes it even more impressive that they were able to convince researchers to rehome these animals

They were reluctant thinking it is not allowed. After presenting the possibilities everyone was happy.

Reviewer 2 Report

Overall, this is a very well-written article. The studies conducted to support the article were well-planned and informative. 

A revision for minor spelling, editing, and grammatical errors in the document are suggested. 

A few edits may be warranted for clarification purposes. 

Define 3R on line 64

Animals (starting on Line 94)- define total n for each evaluation and then overall animal numbers used

Vaccination (starting on Line 118)- information on product details, specifically safety background. These animals were re-homed, but little information on the product they were given was provided. Details on the safety of the plasmid DNA is highly suggested. 

Lines 121 to 134- It is unclear what a 'higher' or 'lower' titer is considered

Lines 150 to 153- It is unclear how IN administration was conducted at the same time the animal was under isoflurane anesthesia. Please explain. 

Overall, this is a very well-written article. The studies conducted to support the article were well-planned and informative. 

Review for minor spelling, editing, and grammatical errors in the document is suggested. 

A few edits are suggested for clarification purposes: 

Define 3R on Line 64.

Animals (starting on Line 94)- define total n for each evaluation and then overall animal numbers used.

Vaccination (starting on Line 118)- information on product details, specifically safety background. These animals were re-homed, but little information on the product they were given was provided. Details on the safety of the plasmid DNA and any exposure in the laboratory (ABSL-2 or -3) environment is highly suggested. 

Lines 121 to 134- It is unclear what a 'higher' or 'lower' titer is considered.

Lines 150 to 153- It is unclear how IN administration was conducted at the same time the animal was under isoflurane anesthesia. Please explain.

ELISA, PsVNA, and VNT – Please include additional details on assay cutoff calculations and limit of detection/lowest detectable limit per initial concentration/dilution used in the assay.

Please use ‘neutralization’ or ‘neutralisation’ consistently.

Line 193 – Include source of Vero E6 cells (and catalog number, if applicable).

Pilot Trial (starting on Line 200) – please include number of males and females for each pilot experiment, please include total volume and/or volume per nostril for each experiment.

Line 224 – Include summary for TCID50 or reference for the process.

Line 225 – Include information on detection limits (what is negative versus positive?) for RT-qPCR.

Reuse and Rehome (starting on Line 255)- Clarification is required for surplus animals rehomed/reused. These were only vaccinated animals (not challenged); however, this is not clear. Please add this information here.

Line 263- Please include the estimated age of animals 'halfway through life expectancy'.

Line 271-272- 19 animals out of how many? May be beneficial to include some details from Figure 1 here.

Lines 281-283- Why was VNT replacement a consideration for viral challenge? These assays have two different goals. It is unclear how VNT replaces a virus titration assay to replace virus challenge. Please include information on use of the VNT for testing of the virus control pilot animal sera (n=18) to confirm the virus has the ability to elicit an antibody response in challenged animals.

Line 301 Figure 3- Please include definition of groups or further details on what is presented in the x-axis.

Lines 301-305- include details on when (timepoint) the serum was collected post-vaccination boost.

Line 308- Titer is based on CPE on the cells. Could this negative result be subjective? Suggest including an image of a comparison of negative wells and positive wells.

Line 309- Suggest adding details to inform the reader that these pilot control animals were not vaccinated (and vaccinated animals were not challenged).

Starting Line 309- How did use of 3 pilot control groups (all n=6) support use of refinement? The animal numbers did not decrease between pilot studies. The vaccinated animals were not challenged and information obtained from these pilot tests (such as clinical signs and days PI for euthanasia) did not apply to the vaccination studies. How exactly were the reasons for refinement, discussed throughout the article, applied to this process- namely the pilot infection and vaccine studies?

Line 358- Define ‘advanced age’.

Line 363- Was information provided to external adopters about the vaccine or basic testing (i.e., animals were not exposed to live virus or exposed to a Biosafety level 3 environment)?

Line 382- It is not clear how the total number of animals on study were reduced. A pilot study examined different parameters than the vaccine study, and no reduction in animal numbers were apparent. Please provide further details.

Line 396 – Should 14 animals read 18 animals (6+6+6 for the pilot study)?

Lines 450-452- How does this information relate to refinement in this study? Refinement using the age of hamsters was not discussion. Please include discussion of the ages of the hamsters used in this study. 

Line 454-472- Discussion of the lack of neutralizing properties for the project was discussed, but no alternatives were suggested. Perhaps a different cell line for the live virus evaluation may have demonstrated a higher antibody response. Notably, no discussion of the lack of antibodies generated by the vaccine itself was discussed, rather than an issue with the assay or animal model. One simple test would be to determine if the challenged hamsters showed a response in the VNT assay. The lack of response may simply be a result of the vaccine properties failing to elicit an immune response in the serum specific to detection of neutralizing antibodies using a live virus, cell-based neutralizing assay.

Line 477- Suggest not including ‘absence of pain’ verbiage. Numerous painful procedures were conducted throughout this testing and it may be misleading to include this statement to describe animal experiences within these studies.

Line 492-493- Suggest including information on cage size for housing 3 hamsters. Aggression of older hamsters is common and was mentioned (Line 496); but it was not clear that ABSL-3 animals, regardless of age, weren’t housed 3 per cage.

Line 530- This statement is true but these examples were not applied for these studies. Suggest including specific examples of how pilot trials in this study were used for refinement.

Line 532-533- It is unclear exactly how VNT can be used to replace virus challenge. Neutralization of antibodies following challenge or vaccine can be determined using a VNT. Challenge of animals can be used as a closed-system to examine the impact of virus in a living model. Please clarify how a VNT can be a replacement for a virus challenge, especially as no neutralizing data were obtained.

Author Response

Define 3R on line 64

OK

Animals (starting on Line 94)- define total n for each evaluation and then overall animal numbers used

added

Vaccination (starting on Line 118)- information on product details, specifically safety background. These animals were re-homed, but little information on the product they were given was provided. Details on the safety of the plasmid DNA is highly suggested. 

Safety testing of the product(s) was not carried out as this would be a later step in the research project. Based on the general DNA plasmid technology, the (in)stability and the time elapsed since the last vaccination (at least 2 months), the designated veterinarian declared the animals safe and suitable for rehoming. Her decision was included in the adoption contract. The exact composition and origin of the products would be confidential – they were described by colleagues in previous work (cited Lainscek et al. 2021); they were produced in a GMP facility and no adverse effects on hamsters were noted. The new owners were given all the information about the procedures the animals were subjected to. In the interest of the readers, we add some information and reference (WHO) on the safety of DNA vaccines under »Reuse and Rehome«

Lines 121 to 134- It is unclear what a 'higher' or 'lower' titer is considered

Explanation added here , it was based on ELISA results and decision making on the team meetings.

Lines 150 to 153- It is unclear how IN administration was conducted at the same time the animal was under isoflurane anesthesia. Please explain. 

Explained. Animal was taken of the mask – grasped for better visualisation, small drops were pippeted on the nostrils and with luck the animal did not awake. If it started to arouse, it would not inhale the res tof the liquid calmly, so it was simply put back on the mask for a minute or two and procedure was repeated.  

ELISA, PsVNA, and VNT – Please include additional details on assay cutoff calculations and limit of detection/lowest detectable limit per initial concentration/dilution used in the assay.

Cut off for each dilution in ELISA (100, 300, 900, 2700, 8100, 24300, 72900, 218700) was calculated from absorbance based on the equation average + standard deviation*2,177 . For a confidence level of 95%, a value of 2,177 was used as the standard deviation multiplier based on the paper of Frey at el.

There is no cut off in pseudovirus system, only the used dilution (50x)

VNT cut off is detectable CPE in Vero E6 monolayer after incubation of 110 h, observed by inverted light microscope at e.g. 400-fold magnification. Characteristic morphological changes in cells (rounding of adherent infected cells) in the culture that were the target of SARS-CoV-2 replication were labeled as CPE.

This information was added in supplement material under corresponding figures. We are adding figures of Vero 6 cells (CPE and control) by request of another reviewer, too.

Please use ‘neutralization’ or ‘neutralisation’ consistently.

Yes a spellcheck mistake, changed to UK type – neutralisation

Line 193 – Include source of Vero E6 cells (and catalog number, if applicable).

done

Pilot Trial (starting on Line 200) – please include number of males and females for each pilot experiment, please include total volume and/or volume per nostril for each experiment.

Edited slightly

Line 224 – Include summary for TCID50 or reference for the process.

The number of infectious viral particles was quantified using the Median Tissue Culture Infectious Dose (TCID50) test. The assay is based on adding a serial dilution of the defined virus to the succeptible cells in 96-well plate. The dilution at which 50% of the wells show CPU is used to mathematically calculate the TCID50 of the virus sample as generally described. Three references were added, but the details are moved to Supplement material as photograpsh are added there, too.

Souf, S. Recent advances in diagnostic testing for viral infections. Biosci. Horizons Int. J. Student Res. 9, (2016).

Pellet, E. P. et al. Basics of virology. Handb. Clin. Neurol. 123, 45-66 (2014).

Lei, C. et al. On the Calculation of TCID50 for Quantitation of Virus Infectivity. Virol. Sin. 36(1), 141-144 (2021).

Line 225 – Include information on detection limits (what is negative versus positive?) for RT-qPCR.

The real-time RT-PCR test was used as a qualitative test. All samples with the fluorescent signal increases exponentially and, producing an exponential curve were treated as positive. All samples were tested with the same procedure. In this study, the LOD was not performed.

Reuse and Rehome (starting on Line 255)- Clarification is required for surplus animals rehomed/reused. These were only vaccinated animals (not challenged); however, this is not clear. Please add this information here.

Ok, thank you for pointing this out.

Line 263- Please include the estimated age of animals 'halfway through life expectancy'.

They were 16 months old at the time of adoption.

Line 271-272- 19 animals out of how many? May be beneficial to include some details from Figure 1 here.

Added

Lines 281-283- Why was VNT replacement a consideration for viral challenge? These assays have two different goals. It is unclear how VNT replaces a virus titration assay to replace virus challenge. Please include information on use of the VNT for testing of the virus control pilot animal sera (n=18) to confirm the virus has the ability to elicit an antibody response in challenged animals.

The viral challenge is of course different test than VNT. To perform VNT, we first prepare a working dilution of the virus in accordance with the general standard. The test serum is then titrated with this standard amount of virus; the procedure applies to all VNT tests (OIE standard). The VNT was not used for pilot animal sera as the time was to short (most animals sacrificed on day 4 post inoculation) and animals could not develop the needed amount of antibodies.

The VNT test was cited as an example of an alternative in vitro method, i.e. as a replacement for viral challenge. It cannot answer all aspects of vaccine efficacy (as we wrote in the introduction) and a viral challenge would still be performed if our vaccine candidate(s) had a prospective efficacy. With all negative VNT results, the viral challenge was obsolete as only neutralising antibodies are considered effective in combating this pandemics. The study of individual cellular immunity or similar was not in scope of the project.

Line 301 Figure 3- Please include definition of groups or further details on what is presented in the x-axis.

Figure 3 moved to Supplement Fig S2, this line was added.

Lines 301-305- include details on when (timepoint) the serum was collected post-vaccination boost.

Added (2 weeks)

Line 308- Titer is based on CPE on the cells. Could this negative result be subjective? Suggest including an image of a comparison of negative wells and positive wells.

Cytopathogenic viruses always induce CPE on infected cells, which is the criterion for determining the result. So if the monolayer of succeptible cells is intact, e.g. no CPE in VNT indicates no apparent viral replication, indicating a negative result. The interpretation of VNT is based on experience, and an inexperienced observer could also give a incorrect result. VNT is used as the gold standard in the diagnosis of many diseases, especially in veterinary medicine.

For clarification we added figures with intact Vero6 cells and two forms of CPE.

Line 309- Suggest adding details to inform the reader that these pilot control animals were not vaccinated (and vaccinated animals were not challenged).

Added

Starting Line 309- How did use of 3 pilot control groups (all n=6) support use of refinement? The animal numbers did not decrease between pilot studies. The vaccinated animals were not challenged and information obtained from these pilot tests (such as clinical signs and days PI for euthanasia) did not apply to the vaccination studies. How exactly were the reasons for refinement, discussed throughout the article, applied to this process- namely the pilot infection and vaccine studies?

Paragraph added here: The use of 6 extra animals (third trial) or pilot trials as whole (18 animals) might contradict the reduction principle of 3R, but was considered as refinement as virulence of specific SARS-CoV-2 strain for golden hamsters was not known. None of the vaccinated animals were subjected to the viral challenge procedure.

More on this topic is in Discussion

Line 358- Define ‘advanced age’.

Over 1 year old

Line 363- Was information provided to external adopters about the vaccine or basic testing (i.e., animals were not exposed to live virus or exposed to a Biosafety level 3 environment)?

Yes, procedures were written in the Adoption contract and explained more in detail if needed during interviews

Line 382- It is not clear how the total number of animals on study were reduced. A pilot study examined different parameters than the vaccine study, and no reduction in animal numbers were apparent. Please provide further details.

The first reduction is considered in the severity of procedures. 40 animals were subjected to vaccination only – a mild procedure, whereas the project proposal expected moderate procedure (also viral challenge). A further reduction was achieved by not taking any further steps in the projects: if the vaccine candidate proved effective, the optimisation of the vaccination protocol would be tested again and the safety of the products would be tested separately, which would require more animals. The broad project proposal also included other animal species (mice and rabbits – the latter for the development of oral mucosal patches, but a detailed discussion of this is beyond the scope of this article). We included also explanation how pilot study (extra animals) might lead to reduction in our case.

Line 396 – Should 14 animals read 18 animals (6+6+6 for the pilot study)?

It should read 12 – initial 12 was planned (6+6), and the third trial was unplanned correction of the viral load inoculum.

Lines 450-452- How does this information relate to refinement in this study? Refinement using the age of hamsters was not discussion. Please include discussion of the ages of the hamsters used in this study. 

A little additional information on the age was added, because it was not our aim to observe the age differences. But when we noticed the sex difference and our hamsters somewhat aged (but were still in their prime) during the vaccination protocols, we realised that many reports have this drawback, as using very young, even prepubescent animals for this kind of translational studies is not exactly beneficial.

Line 454-472- Discussion of the lack of neutralizing properties for the project was discussed, but no alternatives were suggested. Perhaps a different cell line for the live virus evaluation may have demonstrated a higher antibody response. Notably, no discussion of the lack of antibodies generated by the vaccine itself was discussed, rather than an issue with the assay or animal model. One simple test would be to determine if the challenged hamsters showed a response in the VNT assay. The lack of response may simply be a result of the vaccine properties failing to elicit an immune response in the serum specific to detection of neutralizing antibodies using a live virus, cell-based neutralizing assay.

For SARS-CoV-2 replication characteristics and cytopathology, the Vero E6 cell line (African green monkey kidney cells) is commonly used. During test optimisation and validation other animal sera (mice and chicken) were used and researchers claim the test itself was not an issue. The idea to test the serum from challenged hamsters did indeed came up, however, the hamsters were allowed to stay in A-BSL3 for maximum 2 weeks (1 week for acclimatisation) and were not supposed to leave it alive. 1 week is not enough to produce sufficient amount of antibodies. Some of researchers would like to get to the bottom of this but would be another project alltogether. More discussion here would be wild guessing.

Line 477- Suggest not including ‘absence of pain’ verbiage. Numerous painful procedures were conducted throughout this testing and it may be misleading to include this statement to describe animal experiences within these studies.

Absence was changed to »minimising« but definetly there were no »numerous« painful procedures in the study. As instructed by experts in small animal medicine, all painful procedures (blood sampling, i.m. and i.n. vaccination, even ear tagging) were done in general inhalation anesthesia, which also has good analgesic properties in small rodents. I personally inspected all animals for any signs of discomfort after procedures and protocol to eleviate pain was prepared with the designated veterinarian (was anticipated but it did not occur for instance when using complete Freund's adjuvant).

Line 492-493- Suggest including information on cage size for housing 3 hamsters. Aggression of older hamsters is common and was mentioned (Line 496); but it was not clear that ABSL-3 animals, regardless of age, weren’t housed 3 per cage.

In A-BSL3 young animals (second and third trial) were indeed housed 2 or 3 per cage. Animal body mass (below 100 g) and cage dimensions allowed this (2010/63/EU) and as observed the animals preferred to sleep together. Dimensions are added.

Line 530- This statement is true but these examples were not applied for these studies. Suggest including specific examples of how pilot trials in this study were used for refinement.

Added.

Line 532-533- It is unclear exactly how VNT can be used to replace virus challenge. Neutralization of antibodies following challenge or vaccine can be determined using a VNT. Challenge of animals can be used as a closed-system to examine the impact of virus in a living model. Please clarify how a VNT can be a replacement for a virus challenge, especially as no neutralizing data were obtained.

As discussed troughout the paper – with negative result of VNT further testing of vaccinated animals was not justified. Only neutralising effect of the vaccine was the aim of the study.

Reviewer 3 Report

The presented article by Štrbenc et al. describes the development of a laboratory animal model of COVID-19 for vaccine protectivity assessment in SARS-CoV-2 challenge experiments, preclinical immunogenicity assessment of COVID-19 vaccine candidates and the experience of rehoming of golden hamsters following vaccine immunogenicity studies. The studies were performed using adequate methods for vaccine efficacy assessment in a preclinical model: testing of viral RNA levels; testing of anti-RBD/S IgG levels; neutralization assays both in pseudovirus system and with infectious SARS-CoV-2; and histological examination of lung injury.

The title of the paper and the Introduction section state an important ethical problem of possible rehoming of laboratory animals following experimental procedures that allow subsequent living without continuing pain and distress. Undoubtedly, the concept of rehoming implemented by the authors complies with the 3R principles and the described experience is highly valuable.

Still, despite the strong methodological and ethical background of the paper, I must recommend its rejection in the current form with possible resubmission of a completely revised version due to the considerations below.

General comments

1. The data presented Tables 1 and 2, as well as in Figures 2, 3, 4 and 5 are disconnected from the rehoming concept outlined in the title. Generally, the article contains three parts: the development of COVID-19 animal model; immunogenicity assessment of COVID-19 vaccine candidates; and the experience of rehoming, which is the smallest part. 

2. The presented article lacks a measurable and testable hypothesis. The main concept of the paper is illustrated by one figure containing the results of a survey of 11 adopters of golden hamsters, that were previously involved in vaccine immunogenicity assessment experiments. The measurable outcomes of the survey were the opinions of adopters, which do not directly characterize the well-being of the animals and are not adequate to support the hypothesis stated by the authors as "Rehoming as the ultimate refinement of procedures for using golden hamsters in SARS-CoV-2 vaccine research".

Comments about study design

1. The DNA immunization regimens are not explained. Why was the prime 50 mg of DNA and the boost 40 mkg (mistake in text or 1000-fold difference?). Why was the boost done after 4 weeks for animals with higher titers and after 3 weeks for lower titers? Also, only 4-week interval is mentioned in Figure 1.

2. The immunogenicity studies lack a control group, which would be especially valuable for the pseudovirus neutralization assay and ELISA. As stated in the Materials and Methods section, the EPT were calculated based on the pre-vaccination serum, but sera from the control group would be valuable to determine the specificity of the assay. The observed high IgG levels without any neutralization activity is unusual, as mentioned by the authors in Discussion. Moreover, a placebo group which received DNA vector without the insert would definitely be required if the study reached the virus challenge stage.

3. The SARS-CoV-2 challenge in golden hamsters is well described in literature and 10^6 TCID50 is the most widely used dose. Also, the peaks of histological changes are usually observed from Day 4 to Day 7 after challenge. Still, the results of using a different strain (Slovenia/SI -4265/20) and immunohistochemical staining of the spike protein are highly valuable. 

Based on the considerations above I would recommend to publish the positive experience of rehoming of golden hamsters after participation in vaccine immunogenicity studies as a short letter and also to publish the results of virus challenge separately.

Minor issues

line 46. "pandemics" should be replaced by "pandemic"

line 46. COVID-19 is an abbreviation

line 68. not only spike (RBD, pre-fusion, N)

line 70. "The biological assay-virus neutralization test" should be rephrased

lines 72-74. the sentence should be rewritten. What are "some efficacy tests"? The importance of challenge experiments should be described more broadly in this section

line 74. human'e' endpoints. 

lines 88-90. The sentence should be rewritten

lines 130-134. the designation of micrograms should be unified

line 136. protein amount not mentioned

line 167. was blind randomization method used?

line 178. "EPT" abbreviation of endpoint titer was not introduced previously

line 195. Is the presented description correct? Usually for hamster sera the first dilution is 1:8, as lower dilutions are toxic to Vero even with inactivation

line 288. the dose is different from Methods section. mg or µg?

line 287. the phrase "sufficient antibody titers" is misleading, as the levels associated with protection are not defined, thus the sufficient titer is unknown. Also, "IgG titers" would better describe the result

Figure 1. The panels should be named A, B, C, D and better described in the footnote. Is would be better to present the data on the upper two panels in the same way as the data on the lower panels, i.e. with immunization route.

Tables 1 and 2. powers of 10 are mislabeled.

lines 343-345. Was the amount of RNA added to PCR normalized? 18sRNA levels are different between the two experiments

Minor punctuation issues. Mainly double dots, spaces and hyphens instead of dashes.

Author Response

General comments

  1. Indeed most of the co-authors agree there are at least two stories inside this one. However, reserachers performing in-vitro measuremnens would not be willing to publish their results as stand alone paper, seeing the negative results as a failure. This contradicts the need to publish negative results especially when procedures on animals are concerned (discussed in the Discussion). On the other hand LAS specialists can't advise on further procedures without seeing and understanding the results of in-vitro tests. We have followed the suggestion of one of the other Reviewers anf the Editor in charge to present these results in Supplement material. We moved this figures (except Table 2 as it includes clinical scores) to Supplement and added one more figure (asked by another reviewer).
  2. True. Title is changed a bit, the whole paper focuses on all three Rs, most notably on Refinements. The »Rehoming« was an unexpected consequence, not planned , neither was the final survey, but encouraged by institutional wellfare officer and Ethical committee. But still the results deserve the publication as it would be the first documented in this part of EU.

Comments about study design

  1. Yes, of course mkg is an error (I don't see it in the final version?). Third boost was made after 4 weeks in all cases. Animals that initially received 50 mg were given another 50 mg. Animals that had received 20 mg twice received 2 more administrations of 40 mg (double the 20 mg) OR other-product–protein vaccine in two boosts and the period between the third and fourth administrations was 3 weeks. Figure 1 has been altered a little to make this clearer.
  2. Yes, proper control groups were planned (blank DNA vector was delivered, too), but we were limited with the availability of the golden hamsters through Envigo. Instead of the 60 originally ordered, we only received 40, so that not all control groups could be prepared for different protocols. Therefore, we first tried testing different protocols and if any of them seemed to be effective, we would repeat the test (also because Envigo could no longer supply from same breeding facility and we switched to Janvier Labs), redetermine the age, sex and number of the animals and introduce appropriate controls with placebo injections.
  3. Well, at least at beginning (in 2020) we were reading about lower titers and followed those protocols. We did not want our hamsters to suffer too much and now we know better. I agree on the valuability of the data, therfore we ask to be included in Supplement

Based on the considerations above I would recommend to publish the positive experience of rehoming of golden hamsters after participation in vaccine immunogenicity studies as a short letter and also to publish the results of virus challenge separately.

Answer: I follow the recommendation of the Editor and keep it as one paper with supplements.

Minor issues

line 46. "pandemics" should be replaced by "pandemic"

done

line 46. COVID-19 is an abbreviation

changed

line 68. not only spike (RBD, pre-fusion, N)

thank you, added

line 70. "The biological assay-virus neutralization test" should be rephrased

done

lines 72-74. the sentence should be rewritten. What are "some efficacy tests"? The importance of challenge experiments should be described more broadly in this section

this was taken in short from a reference. Rephrased to »initial« eficiency as with already tested products minor changes (adjuvants, dose, combination…) VNT would indeed be sufficient. A sentence added for challenge, but there are more reasons to perform challenge tests, as we focus only on vaccine efficiency here so I don't feel the need (and competence) to diccuss in detail.

line 74. human'e' endpoints. 

yes

lines 88-90. The sentence should be rewritten

Ok

lines 130-134. the designation of micrograms should be unified

some misunderstanding here maybe. Vaccine was calculated to be given in 20 or 50 mg per animal, but was always prepared in 50 µl (different dilutions)

line 136. protein amount not mentioned

it is 100 µg, diluted in 50 µL (added)

line 167. was blind randomization method used?

Kind of – as described. No special randomisation tables were used but with the system set up it worked perfectly. Animals were shifted a couple od times, for procedures they were brought in the second room and this was random.

line 178. "EPT" abbreviation of endpoint titer was not introduced previously

Added, it is now in Supplement

line 195. Is the presented description correct? Usually for hamster sera the first dilution is 1:8, as lower dilutions are toxic to Vero even with inactivation

To my knowledge it was first tried with hthe usual higher dilutions but then repeated many times and with lower dilutions as we could hardly believe the negative results (sera from mice worked fine, and of course the control human serum). This part was written by researcher performing the test.

line 288. the dose is different from Methods section. mg or µg?

well spotted, mg

line 287. the phrase "sufficient antibody titers" is misleading, as the levels associated with protection are not defined, thus the sufficient titer is unknown. Also, "IgG titers" would better describe the result

point taken

Figure 1. The panels should be named A, B, C, D and better described in the footnote. Is would be better to present the data on the upper two panels in the same way as the data on the lower panels, i.e. with immunization route.

We can't agree on labelling as we don't see them as separate pannels. But Figure is corrected and decribed more in detail.

Tables 1 and 2. powers of 10 are mislabeled.

Copy-paste error, corrected

lines 343-345. Was the amount of RNA added to PCR normalized? 18sRNA levels are different between the two experiments

No extra normalisation was used but we followeed the kit protocol. Possibly homogenisation of the tissue was not best the first time. Indeed because of high variance of the 18s RNA taken at second pilot challenge and much better at third we decided to present these values in graph, not only the deltaCT

Round 2

Reviewer 3 Report

The current version of the manuscript addressed most of the concerns of the previous version.

Still, there are a couple of things that need to be corrected:

1. The amount of vaccine used is still different in the text and Supplementary materials — "mg" in line 130 of the manuscript and "ug" in Supplementary Figures.

2. The powers of 10 in the first column of Table 1 are still mislabeled

3. Line 392. Antibody-mediated neutralization is not the only possible mechanism of action of COVID-19 vaccines, i.e. it does not allow to assess the effects of induced T cell response, thus the sentence "the seroneutralisation test or VNT turned out as a complete replacement for the efficacy test with viral challenge of the animals" is misleading. In my opinion this might be further explained in the Discussion section.

4. It is stated in Response that "Instead of the 60 originally ordered, we only received 40, so that not all control groups could be prepared for different protocols. Therefore, we first tried testing different protocols and if any of them seemed to be effective, we would repeat the test (also because Envigo could no longer supply from same breeding facility and we switched to Janvier Labs), redetermine the age, sex and number of the animals and introduce appropriate controls with placebo injections."  As this directly interferes with the Reduction principle, this should be mentioned in Results and discussed.

Minor editing required

Author Response

Thank you for your keen eye.

  1. The micrograms were correct after all, so corrected 3 times in the text.
  2. Found it, corrected
  3. I have added some text here in the discussion (highlighted in yellow). Some researchers did indeed want to test cellular immunity and more, but, to put it bluntly, since the project proposal was narrowly focused on the neutralising properties of the vaccine, we were not at liberty to do anything else (with the animals - due to national strict practise), and later no one was interested in acquiring new funds. I have tried to put this in nice words in the text.
  4. Yellow highlighted text added at beginning of resulty and some explanation later on in the discussion.